Corrected: Author correction

# Room-temperature valley coherence in a polaritonic system

L. Qiu [1,2], C. Chakraborty[2,3], S. Dhara[4] & A.N. Vamivakas [1,2]

The emerging field of valleytronics aims to coherently manipulate an electron and/or hole's valley pseudospin as an information bearing degree of freedom (DOF). Monolayer transition metal dichalcogenides, due to their strongly bound excitons, their degenerate valleys and their seamless interfacing with photons are a promising candidate for room temperature valleytronics. Although the exciton binding energy suggests room temperature valley coherence should be possible, it has been elusive to-date. A potential solution involves the formation of half-light, half-matter cavity polaritons based on 2D material excitons. It has recently been discovered that cavity polaritons can inherit the valley DOF. Here, we demonstrate the room temperature valley coherence of valley-polaritons by embedding a monolayer of tungsten diselenide in a monolithic dielectric cavity. The extra decay path introduced by the exciton-cavity coupling, which is free from decoherence, is the key to room temperature valley coherence preservation. These observations paves the way for practical valleytronic devices.

[1] The Institute of Optics, University of Rochester, Rochester, NY 14627, USA. [2] Center for Coherence & Quantum Optics, University of Rochester, Rochester, NY 14627, USA. [3] Materials Science, University of Rochester, Rochester, NY 14627, USA. [4] Department of Physics, Indian Institute of Technology, Kharagpur 721302, India. Correspondence and requests for materials should be addressed to A.N.V. (email: nick.vamivakas@rochester.edu)

Nanoscale materials have attracted much attention in recent years for their potential to enable optoelectronic device architectures. Among these are monolayer transition metal dichalcogenides (TMDC)[1,2]. Monolayer TMDCs are direct bandgap semiconductors that support stable room-temperature excitons (binding energy of 0.3–0.5 eV). The broken inversion symmetry and strong spin-orbit interaction give rise to pronounced optical selection rules at two energy-degenerate valleys[3–6]. The two valleys, K and K′, can be activated by circularly polarized light with opposite handedness. As a result of the previous, the binary valley pseudospin index has been identified as potential information bearing degree-of-freedom (DOF) giving birth to the field of valleytronics[7,8].

In analogy to spintronics, valleytronics relies on the ability to store, manipulate and readout information from the valley pseudospin. To date, the optical pumping of valley polarization[3–6] and the optical generation, and manipulation of valley coherence have been observed at cryogenic temperatures[9–13]. However, the major obstacle to coherently controlling the valley DOF at room temperature is the intervalley dephasing processes mediated by phonons, the electron-hole interaction or the Maialle-Silva-Sham mechanism (MSS mechanism)[14].

Recent work has demonstrated the potential to realize cavity polaritons based on TMDCs[15–18]. It has been discovered that TMDC polaritons inherit the valley DOF and exhibit enhanced valley polarization at elevated temperature[19–22]. In this work we leverage polaritons based on the monolayer tungsten diselenide (WSe₂) to circumvent intervalley dephasing and preserve finite valley coherence at room temperature. The polariton valley coherence is studied by steady-state angle-resolved Photoluminescence (PL) measurements (See Supplementary Note 1) and explained by a valley-resolved Jaynes–Cummings model[23,24] (See Supplementary Note 2). Our results provide a path to realizing room-temperature valleytronic devices.

## Results

**The polaritonic system.** The device used in this study, Fig. 1a, consists of a $\lambda/2$ cavity (Q-factor 240) formed by a pair of dielectric distributed Bragg reflectors (DBRs) and a monolayer of WSe₂ positioned at the cavity antinode (see Methods). In order to obtain strong coupling the cavity resonance is designed to be commensurate with the WSe₂ neutral exciton energy. However, as shown in Fig. 1b, the exciton peak slightly redshifts from 1.664 to 1.636 eV after SiO₂ deposition likely from strain induced by the deposition process or the dielectric environment change[19], which results in an approximately 10 meV cavity-exciton detuning.

The device dispersion relation is measured by imaging the objective back focal plane onto the entrance slit of an imaging spectrometer[18]. Strong coupling is confirmed by the anti-crossing exhibited by the photonic and excitonic branches, see Fig. 1c. These two branches are well-fit by a coupled-harmonic-oscillator model with an 11.4 meV Rabi splitting (solid lines in Fig. 1c). The cavity and exciton resonant energy obtained from the fitting (dashed lines in Fig. 1c) also agree well with those in Fig. 1b. For the rest of this wok we focus on the lower polariton branch.

**Room-temperature polariton valley coherence.** Next, we study the room-temperature polariton valley coherence. A similar analysis of valley polariton polarization can be found in Supplementary Note 4[19–22]. A schematic of a coherent valley polariton state is illustrated in Fig. 2a. Note that although the polaritons are formed from K (K′) valley excitons, the optically active cavity polaritons lie close to the center of the crystal Brillouin zone. The term valley is used to indicate the K (K′) excitonic component of the cavity polariton. Over-bandgap excitation by linearly

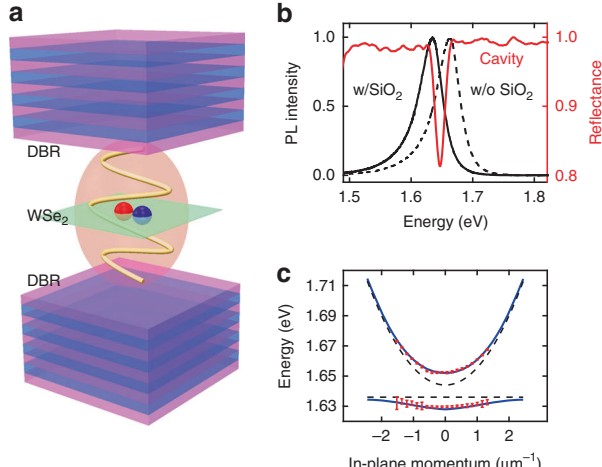

**Fig. 1** Device architecture and optical characterization. **a** Cavity polariton schematic. The green sheet represents monolayer WSe₂. The cavity is assembled by two SiO₂/Ta2O₅ distributed Bragg reflector (DBR) mirrors with a $\lambda/2$ cavity spacing. Strong coupling of excitons and photons leads to the formation of polaritons. **b** Cavity reflectance and WSe₂ PL spectrum. For vertical incidence, cavity resonance is measured as 1.644 eV (red). PL centers at 1.664 and 1.636 eV under ambient conditions before (black dashed) and after (black solid) deposition of the top dielectric layer. **c** Energy-momentum dispersion of cavity polariton. These curves are fitted by a coupled-harmonic-oscillator model with an 11.4 meV Rabi splitting. Dashed lines represent bare resonances and the solid lines stand for two polariton branches. Errorbars are extracted from least square fitting

polarized laser (1.658 eV) first pump the exciton reservoir and coherent superposition of the two valley polariton states is then created through fast relaxation process. The emitted PL passes through a variable angle quarter-wave plate and encounters a linear polarizer before being detected. Figure 2b shows the spectra in the case that the input and output have co- and cross-linear polarization for two different input polarization orientations (no quarter-wave plate). In each panel of Fig. 2c, the integrated PL intensity, as a function of quarter-wave plate angle $\theta$, is fitted to $A \cos (4\theta + \phi) + C_0$ as expected for measuring linear polarization. The fit is used to determine the output polarization angle $\phi_0$. Red arrows in each panel indicate the excitation polarization orientation. The relationship between the measured output linear polarization and the input linear polarization orientation is presented in Fig. 2d revealing aligned polarization orientations as expected for preserved intervalley phase coherence.

To quantify valley coherence the DOLP (degree of linear polarization) $\rho_l$ ($l$ identifies linear) is used which can be expressed as

$$\rho_l = \frac{I_{co} - I_{cross}}{I_{co} + I_{cross}}, \qquad (1)$$

where $I$ is the measured intensity and "co" ("cross") means the input and output linear polarizers are parallel (perpendicular). Figure 3a, b present exemplary co- and cross-polarized PL from the polariton device and a bare WSe₂ monolayer flake. To gain insight into the valley coherence properties (observed in Fig. 3a) and potential depolarization and decoherence mechanisms, we carry out polarization-resolved photoluminescence excitation (PLE) measurements. The results are shown in Fig. 3c (polariton device) and Fig. 3d (bare flake). As the laser excitation energy is swept from far above the WSe₂ bandgap (1.774 eV) to resonance (1.658 eV), $\rho_l$ rises from 0% to ~9% for the polariton device. In contrast (Fig. 3d), the uncoupled exciton from the bare WSe₂

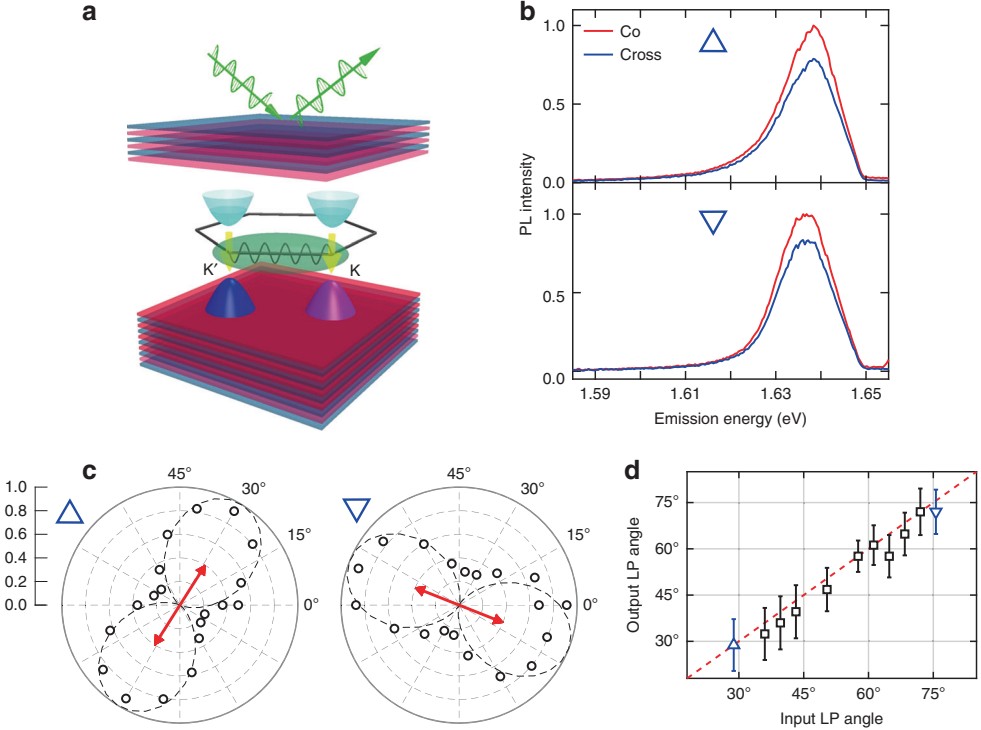

**Fig. 2** Observation of polariton valley coherence. **a** Schematic of coherent valley polariton states. A linearly polarized pump generates a coherent superposition of polaritons arising from the K and K′ valley. The polariton radiatively decays emitting a linearly polarized output. **b** PL spectrum of lower polariton when the excitation and detection polarization are aligned (Co-) and perpendicular (Cross-). **c** Normalized lower polariton PL intensity as a function of detection angle for given input linear polarization (red arrow). The dashed lines are fittings to extract the orientation of output polarization. Baseline (unpolarized part) has been subtracted before normalization. 0.0 and 1.0 stand for the central and outermost contour of the polar diagram, respectively. **d** Relationship between output and input linearly polarized (LP) angles. Dashed line is a linear fit with unity slope, which indicates the lower polariton PL has aligned linear polarization with the pump. Errorbars are extracted from least square fitting

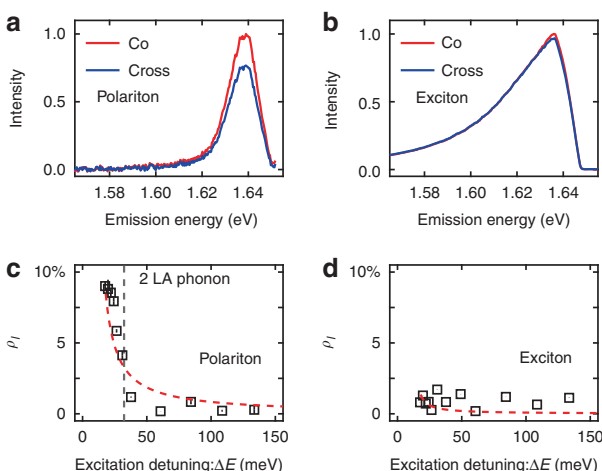

**Fig. 3** Valley coherence of polariton vs. bare exciton. **a**, **b** PL spectrum of lower polariton (**a**) and bare exciton (**b**) under 745 nm linearly polarized pump. Detection and excitation are cross- and co-linearly polarized with each other. **c**, **d** Lower polariton (**c**) and bare exciton (**d**) valley coherence $\rho_l$ as a function of excitation detuning $\Delta E$. Red dashed line is fitted to the same model with MSS dephasing mechanism, gray dashed line is an indication of 2 longitudinal acoustic (LA) phonon energy. Errorbars represent the standard deviation among measurements

monolayer flake shows marginal valley coherence throughout the excitation energy range.

The room-temperature valley coherence preserved in the polariton device, as compared to the uncoupled exciton, is

mainly caused by the photon-exciton hybridization[19–22]. In order to understand the observed polariton photo-physics, we build a valley-specific Jaynes–Cummings model (see Supplementary Note 2). In the strong coupling regime, the polariton valley coherence [see Supplementary Eq. 12], is expressible in terms of the following rates

$$\rho_l = \frac{1}{1 + \frac{\gamma_v + \gamma_{dep}}{\gamma_c + \gamma_x}}, \qquad (2)$$

where $\gamma_c$, $\gamma_x$, $\gamma_v$, and $\gamma_{dep}$ stand for cavity photon relaxation, exciton relaxation, intervalley relaxation, and pure dephasing rates. For comparison, considering bare monolayer WSe$_2$, we can use the same model to predict exciton valley coherence

$$\rho_l = \frac{1}{1 + \frac{\gamma_v + \gamma_{dep}}{\gamma_x}}. \qquad (3)$$

Both equations are of the form $1/(1 + \gamma_1/\gamma_2)$. $\gamma_1$ is the intervalley decoherence rate equal to the sum of $\gamma_v$ and $\gamma_{dep}$. The incoherent intervalley scattering ($\gamma_v$) will lead to decoherence and so will the pure dephasing process ($\gamma_{dep}$). $\gamma_2$ quantifies the population relaxation. In general, valley coherence is the result of competition between population relaxation and coherence decay. If particles (excitons or polaritons) in the two valleys decay to photons before their phase relation decoheres ($\gamma_1 < \gamma_2$), the output DOLP will be preserved. Particularly, by coupling to the cavity mode, polaritons will obtain an extra population relaxation path. This channel is free of decoherence because the cavity mode is immune to any intervalley scattering or pure dephasing processes (see Supplementary Note 2). Consequently, a combination of

accelerated relaxation and suppressed dephasing will boost the polariton's valley coherence.

The PLE results are explained with the previous model. At room temperature, the intervalley scattering is dominated by a phonon-assisted process[4,25–27]. For an exciton transition between valleys, if neglecting the discrepancy between the conduction and valence band dispersion[25], two identical longitudinal acoustic (LA) phonons are needed (one for each the electron and hole). The phonon mode occupation number is expressed as

$$\langle n \rangle = 1/\left(e^{\frac{\hbar\omega_q}{kT}} - 1\right). \quad (4)$$

$\hbar\omega_q$ represents the energy of the phonon modes involved in the process. We identify $kT$ with $\Delta E - \hbar\omega_q$[26], which assumes the excess optical excitation energy $\Delta E$, that is greater than $\hbar\omega_q$ above the exciton energy gap, is assumed to generate phonons. The intervalley transition rate $\gamma_v$ is then proportional to phonon population factor $\langle n \rangle$.

As has already been described theoretically by Maialle et al.[14], the long-range part of the electron-hole interaction will generate a momentum-dependent effective magnetic field, around which the valley pseudospin of excitons with different center-of-mass momentum will precess with different frequencies. This will disturb the phase relation between the two valleys, leading to the following pure dephasing rate:

$$\gamma_{dep} = \frac{1}{2}\langle \Omega^2(k) \rangle \tau_k. \quad (5)$$

The valley dynamics considered here is analogous to the motional narrowing effect observed in a metal or the D'yakonov–Perel (DP) mechanism in semiconductor[28], where the pseudospin relaxation time is directly related to the square of the precession frequency times the momentum relaxation time[29–31]. $\Omega(k)$ is the precession frequency caused by the momentum-related magnetic field, which can be written as $\Omega(k) \approx \sqrt{5}C\alpha(1)|k|/\hbar$. $C$ and $\alpha(1)$ are material-related constants. The center-of-mass momentum $k$ obeys $|k| = \sqrt{2m_e^*\Delta E}/\hbar$, where $m_e^*$ stands for the exciton effective mass and $\Delta E$ is the energy difference between excitation and exciton bandgap. Assuming the momentum relaxation time $\tau_k$ is energy-independent, the dephasing rate $\gamma_{dep}$ is proportional to the energy separation $\Delta E$.

Taken together, Eqs. (4) and (5) allow the energy-dependent valley coherence to be fit as $\rho_l = \rho_{l0}/\left[1 + A_1/\left(e^{\frac{\hbar\omega_q}{\Delta E - \hbar\omega_q}} - 1\right) + A_2\frac{\Delta E}{kT}\right]$. $A_1$ and $A_2$ capture the relative magnitude of the intervalley scattering rate and the pure dephasing rate. The fit agrees well with the data as shown in Fig. 3c. The fitted phonon mode energy is 32.3 ($\pm$2.5) meV, close to the twice of the LA phonon energy at zone edge[32]. The fitting reveals that $A_1/A_2 < 1$, namely, the pure dephasing induced by the MSS mechanism ($\gamma_{dep}$) plays a more important role in the valley coherence dynamics when compared to the intervalley scattering ($\gamma_v$). Additionally, as shown in Fig. 3d, the fit to the bare monolayer exciton data demonstrates the cavity decay rate $\gamma_c$ is much faster than the exciton decay rate $\gamma_x$, which is in good agreement with the fabricated device properties (see Supplementary Note 2). The cavity and exciton linewidths are taken as 6.85 and 0.23 meV[33] respectively, the dephasing time is estimated as 57 fs.

The polariton's behavior is highly sensitive to the excitation laser's angle of incidence due to its dispersive photonic constituent. We investigate polariton valley coherence $\rho_l$ as a function of in-plane momentum $k_{||}$ ($k_{||} = k_0 \sin\theta$ where $k_0$ is the photon wavenumber and $\theta$ is the incident angle). The results are shown in Fig. 4. $\rho_l$ exhibits strong angular dependence, maximizes at $k_{||} = 0$, and then diminishes as the in-plane momentum

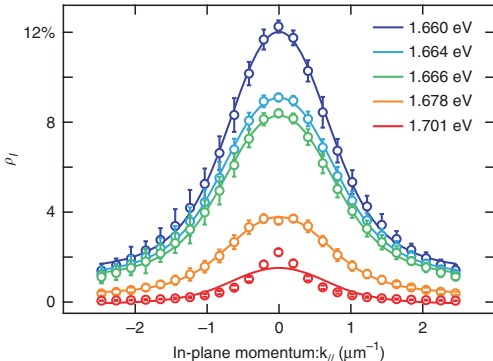

**Fig. 4** Angular dependence of polariton valley coherence. Valley coherence $\rho_l$ at different pump energies are shown as a function of in-plane momentum. Solid lines are fitting results. Errorbars represent the standard deviation among measurements

increases. Also, as the excitation energy is blueshifted from the exciton, the overall magnitude of valley coherence decreases.

Based on the model [see Supplementary Eq. 12] and the device characteristics, the polariton's valley coherence is a function of the cavity-exciton detuning $\delta\omega$. In our case, the cavity and exciton are positively detuned at $k_{||} = 0$, as a result, $\delta\omega$ will monotonically increase as a parabola:

$$\delta\omega(k_{||}) = \delta\omega(0) + \frac{k_{||}^2}{2m_c^*} \quad (6)$$

where $m_c^*$ is the effective mass of cavity photon. The fitting results are shown in Fig. 4. The decrease of the valley contrast is mainly caused by the increasing $\delta\omega(k_{||})$ as a function of in-plane momentum. The overall drop in observed valley coherence is a result in the energy detuning dependence as presented in Fig. 3a.

## Discussion

In summary, our results present evidence for valley coherence of WSe2-cavity polaritons at room temperature. We provide a Jaynes–Cummings inspired model to capture the polariton photo-physics. Important is that although the polariton's excitonic component will undergo depolarization and dephasing processes its photonic counterpart offers a fast population decay channel that preserves the optically accessible valley coherence. We anticipate room-temperature polariton device will provide opportunities for valley-based switching architectures[34–36], as well as provide potential approaches to valley-polaritonic circuitry[37,38]. Future work will focus on the time-resolved measurement of valley polariton devices to provide meaningful insight into polariton depolarization and dephasing dynamics.

## Methods

**Optical measurements**. A Fourier plane spectroscopy system is set-up for the sample under investigation (see Supplementary Note 1). An objective with 0.7 NA is used to focus input light onto the sample under-test, providing an angular range from 0° to 44.4°. Then the back focal plane (Fourier Plane) of the objective is relayed and coupled to the entrance slit of a Princeton (ARC-SP-2758 with PYL-1300BX) spectrometer. Combinations of linear polarizer and quarter-wave plate are inserted in the excitation and collection port. A tunable laser source (Msquared SolsTiS Ti:Sapphire laser) is used for the PLE measurement with power density ~$10^3$ μW μm$^{-2}$. The system is calibrated using a standard gold mirror and background is subtracted to avoid any leakage from the filter.

**Device fabrication**. The DBR mirror is fabricated through physical vapor deposition of alternating layers of SiO2 and Ta2O5, with a λ/4 optical thickness of each layer on a silicon substrate. In order to enhance pumping and collecting efficiency, the top mirror is designed to have fewer pairs than the bottom one. After the deposition of the final Ta2O5 layer, another λ/4 SiO2 is deposited as the half cavity. Monolayer WSe2 is mechanically exfoliated and identified through PL

measurement and optical contrast. Then it is dry-transferred onto the half cavity by a polydimethylsiloxane (PDMS) film. The other half cavity layer is deposited followed by the deposition of the top DBR mirror. Transfer matrix method is utilized to design geometry of the cavity (see Supplementary Note 3).

## Data availability

The data that support the findings of this study are available from the corresponding author upon request.

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

## Acknowledgements

This work was supported by NSF EAGER: 1836566, NSF EFRI EFMA-1542707, NSF CAREER DMR 1553788, AFOSR FA9550-19-1-0074 the Cornell Center for Materials Research with funding from the NSF MRSEC program (DMR- 1719875) and the University of Rochester University Research Award and the Leonard Mandel Faculty Fellowship in Quantum Optics.

## Author contributions

L.Q., C.C., S.D., and A.N.V. conceived the research. L.Q., S.D., and C.C. fabricated the samples. L.Q. and C.C. conducted the measurements. L.Q., C.C., and A.N.V. devised the theoretical model. All authors discussed the data and wrote the manuscript.

## Additional information

**Competing interests:** The authors declare no competing interests.

