## [Peer Review File · Nature Communications]

Reviewers' comments:

Reviewer #1 (Remarks to the Author):

The paper by L. Qiu et al described the results of the investigation the valley coherence in monolayer WSe₂ placed in the microcavity. The main experimental finding is observation of the valley coherence of the polariton in contrast to the bare excitons. The valley coherence is further studied as a function of the excitation energy and these results are explained by Jaynes Cummings model. The preservation of the valley coherence of polariton, in this model, is obtained due to extra relaxation path, which is free of decoherence. The experiments and analysis are performed with care and the main findings are well supported by the experimental results. The results it self could be interesting, but I am not convinced that manuscript should be published in a present form in Nature Communication. In particular, there is another paper (<https://arxiv.org/pdf/1804.09108.pdf>), which presents much more complex and complete investigation of the valley coherence of polariton. I understand that the main novelty presented in the L. Qiu et al is the coherence of the polariton observed at room temperature. However to make the manuscript more complete following points should be addressed:

1. There is a large difference between spectra presented in Fig 2 (a) and 2(b). In fact, the spectrum of bare monolayer has a very asymmetric shape with a long low energy tail. It is not what is usually observed in monolayer WSe₂. Where this shape comes from and why the spectrum changes so dramatically once monolayer is placed in the microcavity?
2. One of the important assumptions of the model is that the main intervally scattering occurs mainly due to phonons at room temperature. In fact mainly due to 2LA phonon. Hence the authors claims that this channel will be very efficient for excitation energy greater than $(2 \hbar \omega_{2LA})$. Did author try to validate it via measurements when ΔE is smaller? Another experimental proof would be temperature dependent measurements. Did author look at the same measurements at low temperature? Such measurements should be performed
3. The exchange interaction, which also leads to the intervalley scattering, has not been taken into account.
4. The data for the upper polariton branch would be also important. Did author performed same measurements and analysis for the upper polariton branch?
5. What happened with the valley coherence as a function of the in plane momentum?
6. Finally, what is a dephasing time authors expect at room temperature?

Reviewer #2 (Remarks to the Author):

The authors report on the elaboration of a microcavity device consisting in an exfoliated WSe₂ monolayer embedded in a $\lambda/2$ -microcavity made of two SiO₂/Ta₂O₅ Bragg mirrors. In this experimental work, the authors demonstrate the possibility of exciton-polariton alignment at room temperature. The authors measurements is in fact a simple manifestation of the polariton pseudo-spin decoherence process between its two Γ -components of opposite helicity during its lifetime, the latter being shortened by the polariton escape process out of the cavity.

Despite the interesting result obtained by the authors, this paper appears to me as mainly incremental, since on one side strong-coupling of exciton-polaritons have already been obtained – although at low temperature indeed – and on the other side exciton alignment is now well documented in the recent literature on monolayer transition metal dichalcogenides (ML-TMDs) at low temperature (see ref. [17] and bibliography below), and in studies performed formerly in wide-gap semiconductor microcavities with quantum wells at room temperature.

About the physics, it is clear that a low-Q microcavity results in short cavity-photon lifetime. Provided strong coupling is maintained, which is made possible due to the strong excitonic oscillator strength – a specificity of ML-TMDs, the resulting polariton escape time from the cavity is shortened with respect to the bare exciton lifetime. As is expected from well-established theory [cf. the review : A. Kavokin and G. Malpuech, Cavity Polaritons (Elsevier, Amsterdam, 2003)], the short polariton lifetime will quench the relaxation of the polariton pseudo-spin and coherence between its two circularly-polarized components.

About the authors' analysis of the effect observed, I have several comments:

1. First, the wording of “Valley coherence” for excitons in TMD is misleading in my opinion, since all the quasi-particles mentioned in this paper (exciton, polaritons) lie in fact close to the centre of the crystal Brillouin zone, *e.g.* in the (polariton)-exciton Γ -valley, as is well documented now in the literature. The valley labelling recalls only the conduction states and the *removed valence states* from which the exciton component of the polariton is made, and which are taken in the *same* K (K') valley for direct excitons. The coherence invoked by the authors occurs between the two Γ -valley polariton states.

The Γ -valley excitons should not be confused with *indirect zone edge excitons* made of conduction and removed valence states of *opposite valleys* (for which polariton cannot be formed indeed) and which centre of mass is close to K (K') points. These are in fact the K -valley excitons.

2 The coherence decay time is basically limited by the exciton polariton lifetime. At zero detuning, taking into account of the cavity Q-factor of $Q \approx 240$, this would lead to a decoherence time in the range 100-200 fs, making difficult any manipulation of coherence during the particle lifetime as required in quantum information processing.

3. The authors argue that the polariton dephasing is dominated by exciton long-range exchange, *i.e.* the Maialle-Silva-Sham mechanism (MSS). This point seems to contradict what

is stated in reference [31] they quote in their own manuscript. As a fact, it is well known that in microcavity devices, the contribution of the TE-TM optical mode splitting to the polariton splitting at $k_{\text{pol}} \neq 0$ is much stronger than the longitudinal-transverse exciton contribution – which rely on long-range Coulomb exchange. The TE-TM splitting effect should even be stronger in TMDs than in QWs due to the strong oscillator strength in these materials.

Moreover, the exciton coherence decay time stated in expression (5) is wrong by a factor 1/2, since according to MSS theory valid for 2D systems it should be written as

$$\gamma_{\text{dep}} = \frac{1}{T_{s2}} = \frac{1}{2} \langle \Omega^2(k) \rangle \tau_k$$
 where τ_k is the exciton linear momentum relaxation time (τ_k is

close to the exciton optical dephasing time measured *e.g.* in four wave mixing experiments).

See for instance the work of M. Glazov *et al.*, (2015) [see ref. below] for ML-TMD excitons.

4. I do not agree with the “intervalley”-scattering mechanism invoked in expression (4). It is clear that, apart from the TE-TM splitting, to change from right to left circular Γ -polariton, this should be done with a two acoustic-phonon process. This should appear as a two-step first order process acting on the exciton component of the polariton, involving necessarily zone edge phonon absorption (hole states should be available). As it is likely that the two involved phonons are different (due to different conduction and valence dispersion close to K-points, the product of the phonon occupation functions should appear as a proportionality factor of this process, instead of expression of formula (4).

5. The model described in the supplement information, relying on Linblad operators for the relaxation processes seems correct. However, it is quite phenomenological and requires the input of many adjustable parameters obscuring the physical meaning of the different process involved.

6. Some important reference on exciton coherence are lacking in the bibliography:

- About exciton alignment manipulations by magnetic field

G. Wang *et al.*, Phys. Rev. Lett. **117**, 187401 (2016)

- About exciton pseudo-spin and alignment relaxation in TMDs :

M. Glazov *et al.*, Phys. Status Solidi B **252**, 2349 (2015)

To conclude, I do not recommend this paper for publication in Nature Communications. I suggest the authors, after revising deeply their manuscript, submit it to *e.g.* Applied Physics Letters, which appears to me more suited for the publication of this interesting, although limited experimental result.

Reviewer #3 (Remarks to the Author):

In their manuscript, Qiu et al. report the observation of valley coherence at room temperature in a microcavity, containing a WSe₂ monolayer in the center. The authors use near resonant linear excitation of the system and observe coherent emission with a polarization degree of up to 9% from the lower energy polariton branch. They analyze their experiments on the basis of a Jaynes Cummings model and find good agreement with their experiments.

The manuscript is clearly written and easy to follow. Valley polarization and valley coherence is currently a hot topic in 2D crystal research. The experiments of Qiu et al. are original and highly interesting for researchers working in the field. I recommend publication in Nature communications.

Response to the reviewers

We thank the reviewers for their thoughtful reading of our manuscript. In the following we address their concerns point-by-point.

Reviewer 1

We are glad to hear the reviewer thinks the experiments and analysis are performed with care and the main findings are well supported by the experimental results. Below we respond in detail to each concern.

Reviewer Point P 1.1 — There is a large difference between spectra presented in Fig 2 (a) and 2(b). In fact, the spectrum of bare monolayer has a very asymmetric shape with a long low energy tail. It is not what is usually observed in monolayer WSe₂. Where this shape comes from and why the spectrum changes so dramatically once monolayer is placed in the microcavity?

Reply: We think what the reviewer means is the difference between Fig 1b and Fig 2b. Firstly, the asymmetric lineshape of WSe₂, as shown in Fig.R1a, is not uncommon in the literature, see [1, 2, 3, 4, 5]. The asymmetric lineshape is also observed in other materials of the TMDC family[6]. The asymmetric lineshape of the exciton peak and the long tail on the side of lower energy can be related to strain[7], exciton-phonon coupling[8] or other excitonic complexes[9]. An additional PL measurement is taken on a fresh-exfoliated monolayer WSe₂ (see Fig.R1b) and a similar asymmetric lineshape is observed. The polariton lineshape, as shown in Fig.R1c, is also asymmetric with a low-energy tail, which is inherited from the exciton. The reduced linewidth is due to coupling with the relatively narrower cavity resonance.

Fig. R1: **WSe₂ Exciton and polariton lineshape.** **a**, Exciton lineshape with Lorentzian fit. **b**, PL spectrum of a new monolayer WSe₂. **c**, Exciton and polariton lineshape.

Reviewer Point P 1.2 — One of the important assumptions of the model is that the main intervally scattering occurs mainly due to phonons at room temperature. In fact mainly due to 2LA phonon. Hence the authors claims that this channel will be very efficient for excitation energy greater than $(2 \hbar \omega_{2LA})$. Did author try to validate it via measurements when ΔE is smaller? Another experimental proof would be temperature dependent measurements. Did author look at the same measurements at low temperature? Such measurements should be performed.

Reply: The data presented in Fig. 3 of the manuscript is shown in Fig.R2c here as a function of ΔE . As a result of the reviewer's concerns we have changed the axis labels of the figures to add clarity to our presentation. Critical to our experiments is the ability to change ΔE such that it is less than twice the LA phonon energy. We do this by changing the laser energy. For clarity, ΔE is the energy difference between the laser excitation energy and exciton bandgap (this has been added to the manuscript after Eq. (4) for clarity). The smallest laser excitation detuning from the exciton gap we can measure in our system, due to technical constraints, is about ~ 20 meV. When ΔE is larger than twice of the LA phonon energy, the dephasing through phonon scattering is efficient and valley coherence is reduced (this reduction is also partially due to the electron-hole exchange interaction pure dephasing with a rate that grows proportional to ΔE in our model). Important in our measurements are the ΔE values less than twice of the LA phonon energy since here coherence is partially restored as the 2LA phonon channel is turned off. We think the data sets we measured as a function of ΔE , in particular for ΔE values less than twice the LA phonon energy as suggested by the reviewer, validate our model.

Fig. R2: **Valley coherence of polariton vs. bare exciton.** **a,b**, PL spectrum of lower polariton (**a**) and bare exciton (**b**) under 745nm linearly polarized pump. Detection and excitation are cross- and co-linearly polarized with each other. **c,d**, Lower polariton (**c**) and bare exciton (**d**) valley coherence ρ_I as a function of excitation detuning ΔE . Dashed curve is fitted to the same model with MSS dephasing mechanism. In **c** the vertical dashed line is the 2 LA phonon energy in WSe₂.

We apologize to the reviewer for some confusion in one of our equations, but there is a typographical error. The excess, laser excited, two-phonon occupation number in our model is not explicitly

temperature dependent (it does not contain kT). The corrected equation is

$$\langle n \rangle = 1 / (e^{\frac{2\hbar\omega_q}{\Delta E - 2\hbar\omega_q}} - 1). \quad (1)$$

Specifically, following the model presented in Kioseoglou et al [10] the excess laser excitation energy above the exciton gap is an additional source of phonons in the system and these excess optically created phonons further spoil the valley-polariton coherence. The maximum valley-polariton coherence we observe is limited by thermal phonons and above gap laser excitation can create excess phonons that further degrade the coherence. In our data, the temperature, which fixes the intrinsic thermal phonon bath, determines the maximum observed valley-polariton coherence, not the structure of the lineshape in the laser energy detuning datasets.

Reviewer Point P 1.3 — The exchange interaction, which also leads to the intervalley scattering, has not been taken into account.

Reply: We agree the exchange interaction can act both on the valley coherence and valley helicity. From the point of valley polarization, it can be interpreted as an annihilation of an exciton in the K (K') valley and the simultaneous generation of an exciton in the K' (K) valley, which will tend to equalize valley population and reduce the observed valley helicity. From the point of valley coherence, this is an incoherent exchange of population between the two valleys. In addition to the previous, excitons with different in-plane momentum will precess between two valleys with different frequencies, which will also lead to dephasing.

Fig. R3: **2LA phonon model.** **a**, Valley helicity (ρ_c) as a function of detuning, red line is fitted to the 2LA phonon model. **b**, Valley coherence (ρ_l) as a function of detuning, red line is fitted to the 2LA phonon model. There is no pure dephasing mediated by the exchange interaction in the model used to generate the dashed fit curve.

However, the reasons we only include the exchange interaction as a dephasing mechanism are as follows:

1. The valley helicity, as shown in Fig.R3a, can be explained by the 2LA phonon model. As a result, we believe phonon-induced scattering dominates the valley depolarization process. For comparison, the data of valley coherence deviates from the 2LA phonon model, as shown in Fig.R3b, which indicates an extra dephasing mechanism is missing.

2. The momentum-dependent precession frequency (from the exchange interaction) between two valleys will act as a pure dephasing mechanism, which will only appear in the time-evolution of the coherences (only affect the off-diagonal elements in the density matrix).

3. The exchange-induced depolarization is more pronounced in the Moly-based materials like MoS_2 and MoSe_2 , see reference[11]. Moly-based TMDCs have a smaller bright exciton energy compared to dark exciton, and Tungsten-based TMDCs have a dark exciton ground state. So the tungsten-based TMDCs are more immune to the exchange-induced processes. The discussion regarding dark excitons is investigated in details in other papers[1, 11, 12] and it is beyond the scope of our model.

Reviewer Point P 1.4 — The data for the upper polariton branch would be also important. Did author performed same measurements and analysis for the upper polariton branch?

Reply: We agree with this point. It is not included because of constraints in our measurement system. When we tune the laser energy close to the exciton, the upper polariton will get overwhelmed by the excitation laser. And if we excite the system far-away from the resonance (1.78 eV) along with suitable long-pass filtering, both the upper and lower polariton branches show negligible polarization response. Below we show data when the laser is tuned to the exciton reservoir (1.641 eV) and a short-pass filter is inserted to get rid of the excitation laser.

Fig. R4: **Upper polariton valley coherence.** **a**, Angle-resolved PL from the upper polariton. The clipping is due to the short-pass filter. **b**, Integrated PL spectrum of the upper polariton when the excitation and detection polarization are aligned (Co-) and perpendicular (Cross-).

As shown in Fig.R4a, part of the upper polariton spectrum close to $k_{//} = 0$ is clipped by the short-pass filter. According to the model, the upper-polariton will preserve the valley coherence better when $k_{//}$ is close to zero. In Figure R4b, the upper polariton's valley coherence is $\sim 4\%$. We didn't present the data of upper polariton branch in the paper because the clipping of the short-pass filter is inevitable, and we miss the valuable information around zero in-plane momentum.

Reviewer Point P 1.5 — What happened with the valley coherence as a function of the in plane momentum?

Reply: In Fig. 4 of the main text, we discuss valley coherence as a function of in-plane momentum. The major angular dependence comes from the cavity dispersion, as shown in Eq. (6) in the main text,

which will result in an angular-dependent exciton-cavity detuning $\delta\omega$

$$\delta\omega(k_{\parallel}) = \delta\omega(0) + \frac{k_{\parallel}^2}{2m_c^*}. \quad (2)$$

Due to the monotonically increasing $\delta\omega$ as a function of in-plane momentum (angle), valley coherence will decrease according to Equation S12 in the supplementary material:

$$\rho_l = \frac{\gamma_c\gamma_x((\gamma_c + \gamma_v + \gamma_x)^2 + 4\delta\omega^2) + (\gamma_c + \gamma_x)(\gamma_c + \gamma_v + \gamma_x)\Omega_{Rabi}^2}{\gamma_c(\gamma_{dep} + \gamma_v + \gamma_x)((\gamma_c + \gamma_v + \gamma_x)^2 + 4\delta\omega^2) + (\gamma_c + \gamma_{dep} + \gamma_v + \gamma_x)(\gamma_c + \gamma_v + \gamma_x)\Omega_{Rabi}^2}. \quad (3)$$

Reviewer Point P 1.6 — Finally, what is a dephasing time authors expect at room temperature?

Reply: The valley coherence we observe at room temperature is 9%, the measured cavity linewidth is 6.85 meV and exciton linewidth is taken as 0.23 meV[13]. As a result the dephasing time (combining pure dephasing and scattering process) is about 57 fs, which is smaller by a factor of two than that of cryogenic temperature measured by Hao Kai, et al[14]. We suspect this difference is mainly due to the temperature. We have added this number into the main text.

Reviewer 2

Reviewer Point P 2.1 — The authors report on the elaboration of a microcavity device consisting in an exfoliated WSE2 monolayer embedded in a $\lambda/2$ -microcavity made of two SiO₂/Ta₂O₅ Bragg mirrors. In this experimental work, the authors demonstrate the possibility of exciton-polariton alignment at room temperature. The authors measurements is in fact a simple manifestation of the polariton pseudo-spin decoherence process between its two Γ -components of opposite helicity during its lifetime, the latter being shortened by the polariton escape process out of the cavity.

Despite the interesting result obtained by the authors, this paper appears to me as mainly incremental, since on one side strong-coupling of exciton-polaritons have already been obtained - although at low temperature indeed - and on the other side exciton alignment is now well documented in the recent literature on monolayer transition metal dichalcogenides (ML-TMDs) at low temperature (see ref. [17] and bibliography below), and in studies performed formerly in wide-gap semiconductor microcavities with quantum wells at room temperature.

About the physics, it is clear that a low-Q microcavity results in short cavity-photon lifetime. Provided strong coupling is maintained, which is made possible due to the strong excitonic oscillator strength - a specificity of ML-TMDs, the resulting polariton escape time from the cavity is shortened with respect to the bare exciton lifetime. As is expected from well- established theory [cf. the review : A. Kavokin and G. Malpuech, Cavity Polaritons (Elsevier, Amsterdam, 2003)], the short polariton lifetime will quench the relaxation of the polariton pseudo-spin and coherence between its two circularly-polarized components.

Reply: We thank the author for their detailed reading of our manuscript. Their many comments and suggestions have greatly improved the quality and clarity of our manuscript. We respond to each specific point below. First, though, we want to emphasize that what we see as the novelty in our device and

experiments is the first demonstration of room temperature coherence for valley-polaritons. There are many attractive features of TMDs and their excitons. One being the potential for room temperature valley-polaritonic devices and crucial for this are demonstrable valley-polarizable and coherent polaritons at room temperature. At the time of our submission, there had been no reports regarding the observation of valley-coherent polaritons with TMD systems (just last month it has been reported at low temperature [15]) and only recently had low temperature and room temperature valley polaritons been observed [16]. We are in complete agreement with the reviewer as to the physical mechanism that make room temperature valley-coherent polaritons observable in our device.

Reviewer Point P 2.2 — First, the wording of "Valley coherence" for excitons in TMD is misleading in my opinion, since all the quasi-particles mentioned in this paper (exciton, polaritons) lie in fact close to the centre of the crystal Brillouin zone, e.g. in the (polariton)-exciton- Γ -valley, as is well documented now in the literature. The valley labelling recalls only the conduction states and the removed valence states from which the exciton component of the polariton is made, and which are taken in the same K (K') valley for direct excitons. The coherence invoked by the authors occurs between the two Γ -valley polariton states. The Γ -valley excitons should not be confused with indirect zone edge excitons made of conduction and removed valence states of opposite valleys (for which polariton cannot be formed indeed) and which centre of mass is close to K (K') points. These are in fact the K- valley excitons.

Reply: We agree with the reviewer's observation that in the two-particle picture the direct excitons arise from the K (K') points electron and holes and the brightest optically active excitons (and polaritons) live at zone center. We use the term valley to capture the excitonic component of the polariton. We apologize for our confused language and have clarified this point by adding the sentence on page 2 linenummer 21: "Note that although the polaritons are formed from K (K') valley excitons, the optically active cavity polaritons lie close to the centre of the crystal Brillouin zone. The term valley is used to indicated the K (K') excitonic component of the cavity polariton."

Reviewer Point P 2.3 — The coherence decay time is basically limited by the exciton polariton lifetime. At zero detuning, taking into account of the cavity Q-factor of $Q \sim 240$, this would lead to a decoherence time in the range 100-200 fs, making difficult any manipulation of coherence during the particle lifetime as required in quantum information processing

Reply: We agree in the present devices exciton-polariton lifetime provide an ultimate limit to possible coherence times. This is a feature of all devices that rely on a system's optically excited states and not peculiar to the cavity polaritons in this work.

Reviewer Point P 2.4 — The authors argue that the polariton dephasing is dominated by exciton long-range exchange, i.e. the Maialle-Silva-Sham mechanism (MSS). This point seems to contradict what is stated in reference [31] they quote in their own manuscript. As a fact, it is well known that in microcavity devices, the contribution of the TE-TM optical mode splitting to the polariton splitting at $k_{\text{pol}} \neq 0$ is much stronger than the longitudinal-transverse exciton contribution which rely on long-range Coulomb exchange. The TE-TM splitting effect should even be stronger in TMDs than in QWs due to the strong oscillator strength in these materials. Moreover, the exciton coherence decay time stated in expression (5) is wrong by a factor 1/2, since according to MSS theory valid for 2D systems it should be written as $\gamma_{dep} = \frac{1}{T_{s2}} = \frac{1}{2} \langle \Omega^2(k) \rangle \tau_k$, where τ_k is

the exciton linear momentum relaxation time (τ_k is close to the exciton optical dephasing time measured e.g. in four wave mixing experiments). See for instance the work of M. Glazov et al., (2015) [see ref. below] for ML-TMD excitons

Reply: Our motivation for reference [31] is to point toward possible applications of room temperature valley-related polariton devices. As the reviewer has noted, there is also theory discussion in that manuscript that is relevant for our study, although we did not initially consult [31] in building our model. Our understanding is that in [31] the MSS mechanism introduces longitudinal-transverse mode splitting, which will result in the TM/TE polariton mode splitting and that description is consistent with our model.

In regards to the TE/TM mode splitting we recognize the large TMD oscillator strengths can result in polariton dephasing. As shown in the figure in Supplementary S3, the TE/TM mode splitting for our cavities is approximately 1 meV, which corresponds to a picosecond dephasing time, much slower than the time scale we are interested here.

Indeed, the large oscillator strength of TMD exciton results in strong mixing of excitonic and photonic states, so TE/TM mode splitting will be transferred to the polariton states. However, the lower polariton is more excitonic-like and we are only interested in the vicinity of the regime where k_{\parallel} is close to zero. We think the long-range electron-hole exchange interaction of exciton will be dominated in this case.

Finally, we appreciate the reviewer catching the missing factor of 2 in the exciton coherence decay time and the factor is corrected in the main text. We have also included a reference to Glazov in the new manuscript.

Reviewer Point P 2.5 — I do not agree with the "intervalley"-scattering mechanism invoked in expression (4). It is clear that, apart from the TE-TM splitting, to change from right to left circular Γ -polariton, this should be done with a two acoustic-phonon process. This should appear as a two-step first order process acting on the exciton component of the polariton, involving necessarily zone edge phonon absorption (hole states should be available). As it is likely that the two involved phonons are different (due to different conduction and valence dispersion close to K-points, the product of the phonon occupation functions should appear as a proportionality factor of this process, instead of expression of formula (4).

Reply:

The model we present is based on that described in Kioseoglou et al [10] (Ref [25] in the main text). In this model, the excess laser excitation energy above the exciton gap is an additional source of phonons in the system and these excess optically created phonons further spoil the valley-polariton coherence. As the reviewer has rightly pointed out (we did not clearly state this in the text and we have made this explicit in the revised version) is the electron and hole bands are symmetric so two equal energy phonons participate in process. In analyzing our data we considered both the simultaneous two LA phonon model and the cascade of two phonons (each assuming equal energy phonons as we wanted to minimize model free parameters[17]). The latter modeling amounts to using the product of phonon occupation numbers instead of twice the LA phonon energy in the phonon occupation number to capture the excess laser generated phonons. In Fig.R5 we present fits to both the valley-polariton coherence and helicity using both models. The top row is the helicity data and the bottom row is the coherence data. Fits to the data reveal that the two LA phonon model, as described in the manuscript, recovers the an energy scale that is commensurate with the K-point LA phonon energy (16.15 meV)[18] in contrast to the cascaded model (21.8 meV). See Fig.R6 for the WSe2 phonon dispersion relation. This being

Fig. R5: **Valley helicity and coherence fittings to different phonon models.** **a,b**, Valley helicity fits to two LA phonon model (**a**) and one LA phonon cascade model (**b**). **c,d**, Valley coherence fits to two LA phonon model (**c**) and one LA phonon cascade model (**d**).

said, we have added a sentence to manuscript to emphasize the assumptions of our model and that it could also be described by a cascade of two unequal energy phonons and that our experiments can't differentiate these processes.

Reviewer Point P 2.6 — The model described in the supplement information, relying on Linblad operators for the relaxation processes seems correct. However, it is quite phenomenological and requires the input of many adjustable parameters obscuring the physical meaning of the different process involved.

Reply: We agree with the reviewer that the developed model is phenomenological in nature. We have attempted to capture the observed steady state features in our experiments by drawing on previous TMD polariton models as well as known sources of dissipation and decoherence. For the valley helicity

Fig. R6: Phonon dispersion curve in monolayer WSe₂[18].

data, the data is fitted by the following equation.

$$\rho_c = \frac{\rho_{c0}}{1 + \frac{A_1}{e^{\frac{E_{phonon}}{\Delta E - E_{phonon}} - 1}}} \quad (4)$$

ρ_{c0} is the "saturated" value of valley helicity, E_{phonon} is the phonon energy and A_1 is the relative depolarization rate, corresponding to $\frac{2\gamma_v}{\gamma_c + \gamma_x}$.

For valley coherence data we have in the main text, it is fitted to Equation.5.

$$\rho_l = \frac{\rho_{l0}}{1 + \frac{A_1}{e^{\frac{E_{phonon}}{\Delta E - E_{phonon}} - 1}} + A_2 \frac{\Delta E}{kT}} \quad (5)$$

A_1 and E_{phonon} are introduced from the results of valley helicity fit with a 5% error. A_2 captures the relative pure dephasing rate $\frac{\gamma_{dep}}{\gamma_c + \gamma_x}$. From our experience of fitting models to data, we do not think there are many free parameters. Also, the helicity fits fix parameters in the coherence fits and the fitting also returns the correct LA phonon energy.

Reviewer Point P 2.7 — Some important reference on exciton coherence are lacking in the bibliography:

- About exciton alignment manipulations by magnetic field: G. Wang et al, Phys. Rev. Lett. 117, 187401 (2016)

- About exciton pseudo-spin and alignment relaxation in TMDs : M. Glazov et al., Phys. Status Solidi B 252, 2349 (2015)

Reply: We have added both references to the manuscript.

Reviewer 3

Reviewer Point P 3.1 — In their manuscript, Qiu et al. report the observation of valley coherence at room temperature in a microcavity, containing a WSe₂ monolayer in the center. The authors use near resonant linear excitation of the system and observe coherent emission with a polarization degree of up to 9% from the lower energy polariton branch. They analyze their experiments on the basis of a Jaynes Cummings model and find good agreement with their experiments.

The manuscript is clearly written and easy to follow. Valley polarization and valley coherence is currently a hot topic in 2D crystal research. The experiments of Qiu et al. are original and highly interesting for researchers working in the field. I recommend publication in Nature communications.

Reply: We are pleased to hear the reviewer found our manuscript interesting and suitable for publication in Nature Communications.

References

- [1] Lundt, N. *et al.* Room-temperature tamm-plasmon exciton-polaritons with a wse 2 monolayer. *Nature communications* **7**, 13328 (2016).
- [2] Yan, T., Qiao, X., Liu, X., Tan, P. & Zhang, X. Photoluminescence properties and exciton dynamics in monolayer wse2. *Applied Physics Letters* **105**, 101901 (2014).
- [3] Mohamed, N. B. *et al.* Evaluation of photoluminescence quantum yield of monolayer wse2 using reference dye of 3-borylbithiophene derivative. *physica status solidi (b)* **254**, 1600563 (2017).
- [4] He, K. *et al.* Tightly bound excitons in monolayer wse 2. *Physical review letters* **113**, 026803 (2014).
- [5] Withers, F. *et al.* Wse2 light-emitting tunneling transistors with enhanced brightness at room temperature. *Nano letters* **15**, 8223–8228 (2015).
- [6] Flatten, L. C. *et al.* Room-temperature exciton-polaritons with two-dimensional ws 2. *Scientific reports* **6**, 33134 (2016).
- [7] Niehues, I. *et al.* Strain control of exciton–phonon coupling in atomically thin semiconductors. *Nano letters* **18**, 1751–1757 (2018).
- [8] Christiansen, D. *et al.* Phonon sidebands in monolayer transition metal dichalcogenides. *Physical review letters* **119**, 187402 (2017).
- [9] Lee, H. S., Kim, M. S., Kim, H. & Lee, Y. H. Identifying multiexcitons in mo s 2 monolayers at room temperature. *Physical Review B* **93**, 140409 (2016).
- [10] Kioseoglou, G., Hanbicki, A. T., Currie, M., Friedman, A. L. & Jonker, B. T. Optical polarization and intervalley scattering in single layers of MoS₂ and MoSe₂. *Sci Rep* **6**, 25041 (2016).
- [11] Baranowski, M. *et al.* Dark excitons and the elusive valley polarization in transition metal dichalcogenides. *2D Materials* **4**, 025016 (2017).
- [12] Zhang, X.-X. *et al.* Magnetic brightening and control of dark excitons in monolayer wse 2. *Nature nanotechnology* **12**, 883 (2017).
- [13] Cui, Q., Ceballos, F., Kumar, N. & Zhao, H. Transient absorption microscopy of monolayer and bulk wse2. *ACS nano* **8**, 2970–2976 (2014).
- [14] Hao, K. *et al.* Direct measurement of exciton valley coherence in monolayer WSe₂. *Nature Physics* **12**, 677–682 (2016).
- [15] Dufferwiel, S. *et al.* Valley coherent exciton-polaritons in a monolayer semiconductor. *Nature communications* **9**, 4797 (2018).
- [16] Sun, Z. *et al.* Optical control of room-temperature valley polaritons. *Nature Photonics* **11**, 491–496 (2017).

- [17] Kioseoglou, G. *et al.* Valley polarization and intervalley scattering in monolayer mos2. *Applied Physics Letters* **101**, 221907 (2012).
- [18] Sengupta, A., Chanana, A. & Mahapatra, S. Phonon scattering limited performance of monolayer MoS₂ and WSe₂ n-MOSFET. *AIP Advances* **5** (2015).

Reviewers' comments:

Reviewer #1 (Remarks to the Author):

I would like to thank the authors for addressing all the points raised by referees. I recommend the manuscript for publication.

Reviewer #2 (Remarks to the Author):

Despite the authors tried to improve the previous manuscripts along with the recommendations of the referees, several points remain unclear in the present manuscript. In my opinion, concerning the excitation spectrum of the linear polarisation of the polariton emission, the authors cannot prove the validity of their interpretation against other possible scenarios which might be plausible as well.

I. My main concern is about the model of Kiseoglu et al. (ref. [26] of the manuscript) which is used here.

First, note that this model was derived for circular emission polarisation (*i.e.* for population relaxation) and at low temperature. Here, we are interested with room temperature experiments and linearly polarised emission (*i.e.* coherence relaxation). We note also that at room temperature, the cavity used by the authors is positively detuned, which means that the lower polariton branch (LP) is mainly of excitonic character, while the upper branch (UP) is mainly of photonic one. As a consequence, the scattering processes of the LB states are only slightly reduced with respect to the excitonic ones.

1. The authors do not really discuss the formation process of exciton polaritons. Are these quasi-particles created in the upper or lower polariton branch? Is the excitation process assisted by optical phonon modes close to Γ point of the Brillouin zone? Do the created particles populate first the polariton “reservoir”, *i.e.* the exciton-polariton states lying out of the light cone, so that they are not coupled to outer photon modes?

2. As far I understand, the model of ref. [26] assumes that, even at low temperature, the dominating process for bright Γ -exciton circular polarisation decay consists, for sufficiently energetic excitons, in emitting a pair of zone edge acoustic phonons, thus changing both the valley index of the electron and of the hole. This means that this process is a sequential one-particle one.

However on one hand, this process appears as strongly unlikely, since the electron states of opposite valleys constituting the bright-excitons of opposite polarisation have *opposite* spin content, so that the exciton pseudo-spin reversal with sequential phonon emission is essentially *forbidden*. As a fact, the electron-phonon interaction cannot change efficiently the spin state (only the orbital one can be modified). A more favourable process with phonons would be to scatter from bright to dark exciton states, located about 40 meV below the bright ones [C. Robert *et al.*, **B 96**, 155423 (2017)].

On the other hand, the exciton polarisation decay process relying on the exciton long-range Coulomb exchange do not suffer from this drawback, and proves very efficient at low temperature (even without taking into account here the TE-TM splitting). As temperature rises, it can enter the sub-picosecond range (Glazov *et al.*, ref. [28] of the manuscript), while the spin/valley reversal for single carriers remains several orders of magnitude longer [Crooker *et al.*, PRL **119**, 137401 (2017), M. Goryca *et al.*, arXiv:1808.01319v1 (2018)].

3. Admitting the sequential process is dominant, it could not preserve valley coherence, since this coherence would be lost immediately after the first one particle scattering process, the

latter transferring the exciton from bright Γ -exciton-polariton state to dark zone edge (K or K') exciton state.

4. The establishment of formula (4), based on a thermodynamically inspired argument, is less than clear. The authors assume that the lattice thermal energy available to operate a valley change for the carriers constituting the LP particle is given by: $kT \equiv \Delta E - 2\hbar\omega_q$, where T is an effective temperature of the considered phonon modes. From this, it can be inferred that:

- i.* The Γ -LP exciton-polaritons are considered to relax quickly to the bottom of their bands.
- ii.* The LP polarisation change by sequential electron-phonon scattering process is made possible only by successive *absorption* of two zone edge phonons, the latter being created by fast *energy transfer* from photo-generated quasi-particles to the lattice.
- iii.* Under stationary conditions, the sequential phonon absorption process is very efficient, so that the energy of the “hot” phonons reservoir is reduced by $2\hbar\omega_q$.

This means implicitly that:

- the zone-edge phonon-modes reach a non-equilibrium population, with a temperature higher than the lattice one T_L , which suppose a very strong phonon emission mode by the photo-generated electronic excitations (*i.e.* more efficient than the phonon lifetime).
- The authors neglect the possible spontaneous emission process which could occur with energetic LP particles, and, for high temperature ($T_L \approx 300$ K), the possible absorption process of zone edge K-phonons which are significantly occupied at the initial lattice temperature T_L .

5. Admitting nevertheless all the previous assumptions, the average occupation of phonon modes $\hbar\omega_q$, would be:

$$\langle n_{\hbar\omega_q} \rangle = \frac{1}{\frac{\hbar\omega_q}{e^{\frac{\Delta E - 2\hbar\omega_q}{k_B T}} - 1}}$$

which supposes a phonon temperature $T = (\Delta E - 2\hbar\omega_q) / k_B$ is reached for the emitted phonon modes. Since the phonons in K and K' are degenerate, the average number of phonon pairs with $q \approx K, q' \approx K'$ would be:

$$\langle n \rangle = \frac{2}{\frac{\hbar\omega_q}{e^{\frac{\Delta E - 2\hbar\omega_q}{k_B T}} - 1}}$$

rather than formula (4). The authors should explain in more details how they get formula (4).

Finally, the authors should have discussed other possible scenarios, as *e.g.* the possible exciton spin relaxation by long-range Coulomb process (see §2). In that case, one may imagine that when the photo-generated particle is excited close to the ground state of the LP polariton, for instance within the range of the acoustic wing of the exciton [S. Shree *et al.*, PRB **98**, 035302 (2018)], the LP escape time competes efficiently with the exchange scattering process, and the linear polarization (“valley-coherence”) could be partly preserved, even at room temperature. However, when the laser excitation energy is increased, the particles are created in the LP reservoir where the coulomb exchange process becomes more efficient (the longitudinal-transverse splitting increases linearly with the LP wave vector),

which leads instead to an efficient depolarisation and decoherence process. The linear polarisation of the emitted light should decrease thus rapidly when the excitation energy increases. This scenario seems to me much more likely, and avoid many questionable assumptions of the previous model.

To conclude this part, too many pending questions are not discussed in the interpretation of the effect observed by the authors, and the model seems unreliable, and even somehow misleading.

II. When the authors claim that the effect observed “*paves the way for practical valleytronic devices*”, one might be quite sceptical.

As a fact, the authors estimate (page 4, second column) themselves that the dephasing time is of the order of 57 fs, so that one may question what can be useful for a practical device operating at room temperature with a so small memory time. We could recall here some of DiVicenzo criterion for quantum information manipulation stating that (see *e.g.* arXiv:quant-ph/0002077v3 13 Apr 2000):

“3. Long relevant decoherence times, much longer than the quantum-gate operation time” is required, so that many quantum manipulations can be achieved during this coherence time.

It should be noted also that the amplitude of the effect is modest, since the linear polarization reached is at most 9% in a quite limited spectral range of excitation detuning (~ 15 meV), so that the process seems not very robust *e.g.* against the spatial fluctuations of the exciton position on the flake due to sample inhomogeneity.

To conclude, I maintain my recommendation for not publishing this paper in Nature Communications. I suggest the authors, after revising deeply their manuscript, submit it to a more applied physics journal, which appears to me more suited for this interesting, although limited experimental result.

Response to Reviewer 2

We thank all the reviewers for their comments and are pleased to learn Reviewer 1 and Reviewer 3 recommend our work for publication in Nature Communications. The comments of all reviewers have improved the manuscript presentation. Before responding to each comment below we want to emphasize that in our density matrix based approach to studying valley-polariton coherence we consider both the 2-phonon intervalley scattering process and long-range Coulomb electron-hole exchange interaction (Maialle-Silva-Sham (MSS) mechanism, main text line 195 -218). And, analysis of the data with our model shows the long-range exchange interaction plays a more important role compared to the phonon-assisted scattering process (main text line 226-230) in describing the steady-state valley-polariton coherence. We think this is aligned with the comments of Reviewer 2 (P2.3 and P2.7 below) and apologize for not making the role of the long-range Coulomb exchange interaction clear in the manuscript. The manuscript has been revised to make certain the previous point regarding the exchange interaction is emphasized. We have included a new figure in the supplementary information (Fig. S7) that demonstrates the importance of the long-range Coulomb exchange interaction in understanding the steady-state degree of valley-polariton coherence.

Below we address Reviewer 2's concerns point-by-point.

Reviewer 2

Reviewer Point P 2.1 — My main concern is about the model of Kiseoglu et al. (ref. [26] of the manuscript) which is used here. First, note that this model was derived for circular emission polarisation (i.e. for population relaxation) and at low temperature. Here, we are interested with room temperature experiments and linearly polarised emission (i.e. coherence relaxation). We note also that at room temperature, the cavity used by the authors is positively detuned, which means that the lower polariton branch (LP) is mainly of excitonic character, while the upper branch (UP) is mainly of photonic one. As a consequence, the scattering processes of the LP states are only slightly reduced with respect to the excitonic ones.

Reply:

We apologize for the confusion and did not clearly state how we used the results of Kiseoglu. Kiseoglu et al. use a rate equation model to predict steady-state exciton population/polarization. Our model is a density matrix approach that allows us to study the steady-state behaviour of both the valley-polariton polarization (population) and valley-polariton coherence (coherence). With the appropriate temperature dependent distribution functions we can model our data at low temperature and room temperature. What we do utilize from the Kiseoglu work is their results on the 2-phonon intervalley scattering rate. Specifically, this rate appears as one of the rates in our density matrix formulation along with the long-range Coulomb exchange interaction. Note, the intervalley phonon scattering process can be treated as an incoherent population transition between two valleys, which will affect both valley-polariton helicity (circular polarization) and coherence (linear polarization).

We are in agreement with the reviewer regarding the cavity detuning. For our device the Hopfield coefficients for the lower polariton at zero in-plane momentum reveal the lower polariton comprises 0.841 excitonic character and 0.541 photonic character. It is true that the lower polariton has a stronger excitonic component, but there is a non-negligible photonic constituent. It is this hybridization that

allows observation of valley-polariton coherence as mentioned previously by the reviewer. And, we agree with the reviewer that the closer we are to resonance the larger enhancement we would observe for the valley-polariton coherence. Nevertheless, in our device we are able to overcome the processes that mask room temperature valley-coherence in individual flakes when we engineer the valley polaritons. Future devices will be designed to enhance valley-polariton coherence and our current device provides a proof-of-principle that room temperature valley-coherent polaritons are possible.

Reviewer Point P 2.2 — The authors do not really discuss the formation process of exciton polaritons. Are these quasi-particles created in the upper or lower polariton branch? Is the excitation process assisted by optical phonon modes close to Γ point of the Brillouin zone? Do the created particles populate first the polariton reservoir, i.e. the exciton-polariton states lying out of the light cone, so that they are not coupled to outer photon modes?

Reply: We regret not being clear on this point. Depending on the excitation energy, absorption occurs either directly into excitons (for high energy) or into polaritons and is mediated by the excitonic component. All excitation occurs within the light cone. As the particles relax down the polariton bands a non-thermal phonon reservoir is created. Figure R1 illustrates the excitation mechanism. The focused laser excites the system at a specific energy across a range of in-plane momenta determined by the focusing objective numerical aperture; all which lie within the light cone. This figure is added to the supplementary information for clarity.

Fig. R1: **Illustration of laser excitation** .

Reviewer Point P 2.3 — As far I understand, the model of ref. [26] assumes that, even at low temperature, the dominating process for bright Γ -exciton circular polarisation decay consists, for sufficiently energetic excitons, in emitting a pair of zone edge acoustic phonons, thus changing both the valley index of the electron and of the hole. This means that this process is a sequential oneparticle one. However on one hand, this process appears as strongly unlikely, since the electron states of opposite valleys constituting the bright-excitons of opposite polarisation have opposite spin content, so that the exciton pseudo-spin reversal with sequential phonon emission is essentially forbidden. As a fact, the electron-phonon interaction cannot change efficiently the spin state (only

the orbital one can be modified). A more favourable process with phonons would be to scatter from bright to dark exciton states, located about 40 meV below the bright ones [C. Robert et al., B 96, 155423 (2017)]. On the other hand, the exciton polarisation decay process relying on the exciton long-range Coulomb exchange do not suffer from this drawback, and proves very efficient at low temperature (even without taking into account here the TE-TM splitting). As temperature rises, it can enter the sub-picosecond range (Glazov et al., ref. [28] of the manuscript), while the spin/valley reversal for single carriers remains several orders of magnitude longer [Crooker et al., PRL 119, 137401 (2017), M. Goryca et al., arXiv:1808.01319v1 (2018)].

Reply: First, regarding conservation rules, we agree that the phonon-assisted process must conserve spin and a bright-to-bright intervalley exciton transition will require the simultaneous spin reversal of an electron and hole. As the reviewer previously commented in the first round of reviews “to change from right to left circular Γ -polariton, this should be done with a two acoustic-phonon process.” And, as stated in [27], “Note that since a spin-flip during a single intervalley scattering event is unlikely in most known materials, multiple intervalley scattering processes will be required to account for the simultaneous spin flip of electrons and holes.” And, as the authors of [27] go on to discuss “there are two mechanisms that could be responsible for the electron or hole spin-flip during this phonon mediated intervalley scattering event. One is that the spin-flip is mediated by short range scattering from impurities. The presence of a background carrier population could enhance the probability of such a process[1]. The other mechanism is that intervalley scattering proceeds through the nearly spin-degenerate Γ -valley of the Brillouin zone[2].” The mentioned references being:

[1] Song, Y. and Dery, H. Transport theory of monolayer transition-metal dichalcogenides through symmetry. Phys. Rev. Lett. 111, 026601 (2013).

[2] Mai, C. et al. Many-body effects in valleytronics: Direct measurement of valley lifetimes in single-layer MoS₂. Nano Lett. 14, 202-206 (2014).

With the previous in mind it is possible for the 2-phonon intervalley scattering to respect all conservation rules.

Next, regarding the use of a 2-phonon intervalley scattering rate and the long-range Coulomb interaction. Our model of the steady-state valley-polariton coherence consists of two pieces. One piece is an incoherent population transfer mediated by a 2-phonon intervalley scattering and the other piece is a pure dephasing that is a manifestation of the MSS long-range Coulomb electron hole exchange interaction. Both of these rates are incorporated into our density matrix model that yields the steady-state expressions for the valley-polariton polarization and coherence. The reason we have used the 2-phonon intervalley scattering rate in our model is that in the context of valley polarization decay, it is well established that 2-phonon intervalley scattering is the dominant processes. This is reported in our references [25] and [27] as well as in Zeng, et al *Valley polarization in MoS₂ monolayers by optical pumping* Nat. Nanotechnol. 7, 490-493 (2012). We also demonstrate this is the case in our Supplementary Information for valley-polarized polaritons (see Figure S5 in the Supplementary Information). And, more recently, a double Resonance Raman study was reported in Carvalho, B. R. et al. *Intervalley scattering by acoustic phonons in two-dimensional MoS₂ revealed by double-resonance Raman spectroscopy* Nat. Commun. 8, 14670 (2017) where there is extensive discussion regarding the role of acoustic phonons in valley depolarization. It can be found in the paragraph just preceding the Discussion section on page 7. Specifically: “Our conclusion that the LA(K) phonon dominates the intervalley scattering is consistent with previous reports of valley depolarization in TMDs...”

And they concluded (on p. 7 in the last paragraph of their Discussion section):

“Finally, we show the second-order DRR spectra of MoS₂ originates in intervalley scattering by acoustic

phonons, a mechanism which is also responsible for the destruction of valley polarization (that is, depolarization) [41,43]. Our work is thus relevant for the field of valleytronics of MoS₂, since the robustness of valley polarization depends sensitively on the absence of intervalley scattering [41,42,45].” Note their reference [41] is Zeng, et al *Valley polarization in MoS₂ monolayers by optical pumping* Nat. Nanotechnol. 7, 490-493 (2012) and their reference [43] is our reference [27].

Since incoherent population transfer will also lead to loss of coherence it is natural to incorporate this mechanism in our model of valley-polariton coherence. And from these prior works it is not unexpected that 2-phonon intervalley scattering should play a role. We think possibly our discussion surrounding the role of acoustic phonons in mediating the intervalley scattering was confusing so we have carefully reworded this section to improve clarity.

And, we completely agree with the reviewer on the importance of the long-range Coulomb exchange interaction. Our model returns a steady-state degree of linear polarization (a proxy for valley-polariton coherence) that requires not only 2-phonon scattering (incoherent population transfer between the two polariton branches), but also pure dephasing mediated by the long-range electron-hole exchange interaction (MSS mechanism, main text line 195 -218). Our model captures both the intervalley scattering and long-range exchange interaction, and the fitting result indicates the long-range interaction plays the more important role in the valley decoherence process. In Fig. R2 we show how our models fits to the valley-polariton coherence data when we remove the long range Coulomb interaction pure dephasing. It is clear from Fig. R2 that the long-range Coulomb exchange interaction is crucial to explain our data. Note: The reference [26] mentioned by the reviewer is updated as reference [27].

Fig. R2: **Steady-state valley-polariton coherence** Steady-state valley-polariton coherence (ρ_l) as a function of laser excitation detuning from the gap. Red dashed line is fitted to our model, but there is no pure dephasing mediated by the long range Coulomb exchange interaction. The model does not fit well in this case.

Reviewer Point P 2.4 — Admitting the sequential process is dominant, it could not preserve valley coherence, since this coherence would be lost immediately after the first one particle scattering process, the latter transferring the exciton from bright Γ -exciton-polariton state to dark zone edge (K or K') exciton state.

Reply: We agree with the reviewer. It is true that the phonon-assisted intervalley scattering will not preserve valley coherence, since it results in an incoherent population transfer. This is one mechanism that reduces the valley-polariton coherence, the other is the long-range Coulomb exchange interaction.

Reviewer Point P 2.5 — The establishment of formula (4), based on a thermodynamically inspired argument, is less than clear. The authors assume that the lattice thermal energy available to operate a valley change for the carriers constituting the LP particle is given by: $kT = \Delta E - 2\hbar\omega_q$, where T is an effective temperature of the considered phonon modes. From this, it can be inferred that:

- i. The Γ - LP exciton-polaritons are considered to relax quickly to the bottom of their bands.
- ii. The LP polarisation change by sequential electron-phonon scattering process is made possible only by successive absorption of two zone edge phonons, the latter being created by fast energy transfer from photo-generated quasi-particles to the lattice.
- iii. Under stationary conditions, the sequential phonon absorption process is very efficient, so that the energy of the “hot” phonons reservoir is reduced by $2\hbar\omega_q$.

This means implicitly that:

- the zone-edge phonon-modes reach a non-equilibrium population, with a temperature higher than the lattice one T_L , which suppose a very strong phonon emission mode by the photo-generated electronic excitations (i.e. more efficient than the phonon lifetime).

- The authors neglect the possible spontaneous emission process which could occur with energetic LP particles, and, for high temperature ($T_L > 300$ K), the possible absorption process of zone edge K-phonons which are significantly occupied at the initial lattice temperature T_L .

Reply: Please see the response after P2.3. We think including an intervalley scattering that is mediated by a laser induced nonequilibrium acoustic phonon population is not only appropriate due to prior work, but also results in a fit to our data that returns a meaningful energy for the WSe_2 LA phonon. There is a clear increase in the steady-state valley-polariton coherence once the excitation's energy detuning is less than the energy of two WSe_2 LA phonons. For this reason we think our model is reasonable and does not need to include other processes to capture the behaviour of our data.

Reviewer Point P 2.6 — Admitting nevertheless all the previous assumptions, the average occupation of phonon modes $\hbar\omega_q$, would be:

$$\langle n_{\hbar\omega_q} \rangle = \frac{1}{e^{\frac{\hbar\omega_q}{\Delta E - 2\hbar\omega_q}} - 1}$$

which supposes a phonon temperature (2) is reached for the emitted phonon modes. Since the phonons in K and K' are degenerate, the average number of phonon pairs with $q \sim K$, $q' \sim K'$ would be:

$$\langle n \rangle = \frac{2}{e^{\frac{\hbar\omega_q}{\Delta E - \hbar\omega_q}} - 1}$$

rather than formula (4). The authors should explain in more details how they get formula (4).

Reply: As the reviewer mentions, the intervalley phonon scattering rate is proportional to the phonon occupation factor $\langle n \rangle = \frac{1}{e^{\frac{\hbar\omega_q}{kT}} - 1}$. We identify kT with $\Delta E - \hbar\omega_q$ and express the phonon occupation

factor a $\langle n \rangle = \frac{1}{\frac{\hbar\omega_q}{\Delta E - \hbar\omega_q} - 1}$ where ΔE is the excess excitation energy above the energy gap. As mentioned previously, this excess energy will create a non-thermal distribution of acoustic phonons. When we fit our data to both the steady-state valley-polariton polarization and steady-state valley-polariton coherence models we find a value for $\hbar\omega_q$ that is equal to two times the energy of the K point longitudinal acoustic phonon in WSe₂. This is consistent with previous reports (from other TMDCs) and expected since (see P2.3) two phonons are required for the intervalley scattering process. Note in the fits to the steady-state valley-polariton coherence data we also include the long-range Coulomb exchange interaction. Regarding the factor of 2, consulting the equation in P2.7, when we fit to our data this 2 will be absorbed in to the fit parameter.

We agree the language of phonon pair function may be confusing, and as suggested by the reviewer we have modified the text around lines 187-193 to reflect the above discussion.

Reviewer Point P 2.7 — Finally, the authors should have discussed other possible scenarios, as e.g. the possible exciton spin relaxation by long-range Coulomb process (see §2). In that case, one may imagine that when the photo-generated particle is excited close to the ground state of the LP polariton, for instance within the range of the acoustic wing of the exciton [S. Shree et al., PRB 98, 035302 (2018)], the LP escape time competes efficiently with the exchange scattering process, and the linear polarization (“valley-coherence”) could be partly preserved, even at room temperature. However, when the laser excitation energy is increased, the particles are created in the LP reservoir where the coulomb exchange process becomes more efficient (the longitudinal-transverse splitting increases linearly with the LP wave vector), which leads instead to an efficient depolarisation and decoherence process. The linear polarization of the emitted light should decrease thus rapidly when the excitation energy increases. This scenario seems to me much more likely, and avoid many questionable assumptions of the previous model.

Reply: As we have described in previous comments, the Maialle-Silva-Sham mechanism (long-range electron-hole interaction) is the dominating dephasing mechanism (line 226-230 in the main text). As we fit our energy-dependent valley-polariton coherence data by the following equation (line 221 in the main text; edited in the revised version for clarity)

$$\rho_l = \frac{\rho_{l0}}{1 + A_1 \frac{1}{\frac{\hbar\omega_q}{\Delta E - \hbar\omega_q} - 1} + A_2 \frac{\Delta E}{kT}}$$

The term leading by A_1 in the denominator stands for the phonon-assisted scattering process; while the term leading by A_2 represents the long-range exchange interaction. In fact, as mentioned in the maintext (line 226-230), $A_1 < A_2$, which suggests the dominating process is the long-range exchange interaction.

Reviewer Point P 2.8 — When the authors claim that the effect observed “paves the way for practical valleytronic devices”, one might be quite sceptical.

As a fact, the authors estimate (page 4, second column) themselves that the dephasing time is of the order of 57 fs, so that one may question what can be useful for a practical device operating at room temperature with a so small memory time. We could recall here some of DiVicenzo criterion for quantum information manipulation stating that (see e.g. arXiv:quantph/0002077v3 13 Apr 2000)

Long relevant decoherence times, much longer than the quantum gate operation time is required, so that many quantum manipulations can be achieved during this coherence time.

It should be noted also that the amplitude of the effect is modest, since the linear polarization reached is at most 9% in a quite limited spectral range of excitation detuning (15 meV), so that the process seems not very robust e.g. against the spatial fluctuations of the exciton position on the flake due to sample inhomogeneity.

Reply: We agree that the dephasing time is short for the current device, but there is plenty of room for future optimization. As the reviewer mentioned in the first comment, a more resonant cavity will have a larger photonic constituent and will offer an opportunity to enhance the room temperature coherence contrast. Important, our work is the first proof-of-principle device demonstrating the possibility of room temperature coherence.

REVIEWERS' COMMENTS:

Reviewer #1 (Remarks to the Author):

First I would like to thank the authors for addressing and clarifying the extensive comments of the second referee. In my opinion all the points have been very clearly explained by authors and I am still supportive for publications.